# Belonging in Online Synchronous Classrooms: Experiences of Minoritized Students in Dutch Higher Education during the COVID-19 Pandemic

**Tisja Korthals Altes** [1] , **Seda Muftugil-Yalcin** [2,*] and **Marieke Slootman** [3]

1   Teachers and Teacher Educators in Inclusive Learning Environments, Windesheim University of Applied Sciences, 8017 CA Zwolle, The Netherlands; t.korthals.altes@windesheim.nl
2   Department of Organization Sciences, Vrije Universiteit Amsterdam, 1081 HV Amsterdam, The Netherlands
3   Youth and Society, Research Group, Inholland University of Applied Sciences, 1081 HV Amsterdam, The Netherlands; m.w.slootman@vu.nl
*   Correspondence: seda.yalcin@vu.nl

**Abstract:** In response to the COVID-19 pandemic, higher education in many countries suddenly moved online. The sudden move to online education forced teachers to adapt their teaching to an online context. The concept of belonging has been described as the next evolution of, and a characteristic that is central to, inclusion. Many have argued that inclusion can never be truly achieved without the presence of belonging. In this paper we ask important questions, such as how does the move to online education, with its lack of personal cues and heightened anonymity, influence minoritized students' sense of belonging? What is the role of the sense of belonging for inclusion in education? From our data about the minoritized student experiences of belonging in online classrooms, three patterns emerged. (1) The most dominant experience was that the lack of connection in online classrooms reduced students' sense of belonging. (2) In a few instances, however, the lack of connection and the heightened anonymity of online classrooms led to an increased sense of belonging and safety in online classrooms. (3) However, not all students experienced the switch to online education as a major change.

**Keywords:** online education; minoritized students; inclusion in education; sense of belonging; higher education





## 1. Introduction

Sense of belonging is broadly concerned with individual perceptions of feeling connected to and cared for by others [1,2]. There are many authors who argue that belonging is a fundamental human need and that individual perceptions of belonging can result in both negative and positive effects [3–6]. Building on the research by Shore and colleagues (2011), we believe that inclusion can be conceptualized as "the degree to which an individual perceives that the group provides him or her with a sense of belonging and authenticity" [7] (p. 373).

The centrality of a sense of "belonging" to the inclusion process in schools is also being increasingly recognized, with Warnock (2005) arguing that "The concept of inclusion must embrace a sense of belonging, since such a feeling appears to be necessary both for successful learning and for more general wellbeing" [8] (p. 15) [9]. In the university context, a positive sense of belonging aims for inclusive education.

The COVID-19 pandemic caused the closure of university campuses around the world, leading to disruptive and rapid changes in learning and teaching. It instigated what has been described as an "enforced online migration" [10] (p. 9), where all learning, teaching and assessment moved to online domains. Prior to this shift, as argued by Gravett and Ajjawi (2021) "belonging" was generally contextualized through extracurricular activities

(e.g., sports, culture, clubs), curricular belonging (e.g., engagement, peer friendship) and also the physical places and spaces of university campuses [11] (p. 1386). With the advent of COVID-19, however, these taken-for-granted spaces and places of belonging on campus have been displaced, and educators are increasingly preoccupied with questions pertaining to how to engage their students, how to foster a sense of community and how to develop a sense of belonging within online and hybrid learning contexts [11].

For many educators, sense of belonging is a valued concept in campus-based learning, being firmly linked with improved student attainment, increased learner satisfaction and reduced attrition rates [12]. They believe that the need to belong is a fundamental human desire that provides the basis for relationships and community and that this sense of belonging is pivotal to university students [13]. Amidst all the scholarly debates around what this new online and hybrid education meant in terms of "sense of belonging", a new line of discussion emerged about the prospect of online education for "inclusive education". Scholars emerged who highlighted how online learning can offer benefits, for instance, in creating accessible and inclusive classrooms. Regarding inclusive classrooms, what is generally meant is a classroom in which every student has the equal ability to experience belonging in the classroom and to successfully finish the course, regardless of their identity, background, body, and physical or neurological abilities. Farrell and Brunton (2020) point to accessibility, when they state that "Online learning is the fastest growing area of education worldwide because it provides access to educational opportunities in a flexible manner to students from diverse backgrounds and geographical regions who often cannot access higher education by other means" [14] (p. 1). Blake (2000) and Sullivan (2002) argue that online courses can be more accessible and more inclusive for a diverse student population than offline courses due to their anonymity, as this can reduce stereotyping [15,16]. Peacock et al. (2020) add that a decreased sense of hierarchy can strengthen the student–instructor connection in online classrooms [12].

In 2014, Delahunty, Verenikina and Jones, explained that "in terms of online pedagogy, rapidly changing technologies have outpaced research on how to appropriately address the intangible social space of the virtual classroom" [17] (p. 244). This situation has significantly worsened in view of the acceleration of change instigated by the COVID-19 pandemic, and as such, many educators are seeking further information about how students experience online learning. We are aware that the digitalization of education will continue due to its benefits in terms of flexibility [18]. Because online education is here to stay, we consider it essential to better understand how minoritized students in particular experience online education. We ask: How does the move to online education, with its lack of personal cues and heightened anonymity, influence minoritized students' sense of belonging? What is the role of belonging for inclusion, particularly among minoritized students?

We provide an answer to these questions based on qualitative interviews with minoritized students who were studying at the Vrije Universiteit Amsterdam during the COVID-19 pandemic. In March 2020, the Dutch government prohibited large gatherings, including those for higher education purposes. The sudden move to online education forced teachers to adapt their teaching to an online context. In most courses at the VU at that time, on-campus courses had been replaced primarily by online synchronous classrooms that made use of an online conference tool, such as Zoom or Teams, sometimes supplemented by short online asynchronous presentations. Before we present the results, we first reflect on belonging, study success and online education in relation to debates on "inclusive education" based on the existing literature and describe our methodological approach. We finish the paper with a discussion of the relevance of our findings and with practical recommendations for educational institutions.

## 2. Theory: The Relevance of Belonging in Online Education

A sense of belonging is important for study success, both offline and online. A sense of belonging is an individual's subjective feeling of fitting in with a group and/or environment, which connects to a sense of connection and safety. Belonging goes further

than "feeling at home": it is the expectation of being valued as a rightful participant of the group [19,20]. Belonging is crucial for study success, as Tinto (1993) explains. He argues that student retention depends on "academic integration" (which includes positive relationships with educators and course content) and "social integration" (which includes positive relationships with peers) [21]. Thomas, Herbert and Teras (2014) emphasize the importance of a sense of belonging for retention in online courses specifically. They show that similar to offline courses, students are more likely to complete their online courses when they feel part of a student community [22]. Freeman, Anderman and Jones (2007) observed that students' sense of belonging in the classroom is connected to their sense of belonging at their university [23]. Belonging influences student engagement [24], which in turn enhances "deep learning". Higher education researchers Entwistle and Waterston (1988) [25], Barkley (2010) [26] and Barnett (2015) [27] state that deep learning (critical thinking and deeper understanding) is one of the main goals of higher education and deep learning should thus be facilitated both in online and offline higher education.

Minoritized students generally experience lower levels of belonging in higher education [19]. They more often experience exclusion and microaggressions (like stereotyping, spotlighting or jokes) [28,29], and those who are not raised in white, middle-class environments are relatively unaccustomed to the norms and codes that are dominant in higher education (to the academic habitus) [30]. Heterosexism and cisgenderism render diverse sexual and gender identities non-normative and, hence, stigmatized [31]. In the school environment, such ideologies are linked to a lower sense of belonging for lesbian, gay, bisexual, transgender or queer (LGBTQ) students when compared to their heterosexual and cisgender peers [32,33].

In this paper, we prefer to use the term "minoritized" rather than using terms such as "minorities", "members of minority groups" or "underrepresented minorities", because using "minoritized" makes it clear that being minoritized is about power and equity rather than numbers. It gives us a powerful way to promote racial justice by appealing to the common experience of being excluded. We are aware that while using "minoritized" we risk creating a false equivalence that sees all instances of being minoritized as equal and discounting unique forms of oppression by subsuming them under a single term, but we still think that using this term carefully can ensure that its advantages outweigh these risks [34].

Online education in general is said to present challenges for belonging and engagement. Farrell and Brunton (2020) give an extensive overview of the literature that describes these challenges [14]. Woodley and Simpson (2014) show that online degree programs have lower completion rates in comparison with offline programs [35]. Among the many reasons for student attrition are a lack of student engagement [36,37]. This has to do with the many social cues, like body language and facial expressions that display emotion, being less clear online [38,39], which makes it hard to establish personal connections online [40]. Yet warm, respectful interactions strongly contribute to belonging in classrooms [23]. In other words, in online environments, it is harder to establish a "social presence", which is the ability to establish social and emotional connections, and to present oneself as a "real person" to group members (see Community of Inquiry Framework, 2022 [41]).

As mentioned, despite these challenges in terms of belonging and social presence, online education is sometimes seen as promising for minoritized students. Besides the enhanced opportunities for accessibility because of the flexibility that online education can provide [14], others point to the opportunity to create a safe environment for minoritized students, because the lack of social presence and the accompanying anonymity can reduce stereotyping [15,16,42], and the decreased sense of hierarchy can strengthen the student–instructor connection [12]. In addition, for students who feel excluded from the dominant student community in offline education, the absence of a strong student community in online education can come as a relief. However, as Ramdas et al. (2019) suggest that the absence of a student community can also lead to feelings of unsafety [43].

The concept of belonging has been described as the next evolution of, and a characteristic that is central to, inclusion [44,45]. Many have argued that inclusion can never be truly achieved without the presence of belonging. Falvey and Givner (2005) wrote that the underlying assumption of inclusive education "is about embracing everyone and making a commitment to provide each student in the community, each citizen in a democracy, with the inalienable right to belong" [46] (p. 5). Inclusion thus refers to the design of educational environments to take account of the needs of all learners, providing access to a learning experience at a pace and level to suit the learner.

We are also very well aware that the dominant conceptualizations of belonging in higher education often contain a number of key omissions because this concept and its adoption is not straightforward as it might appear at a first glance. The first omission is related to how belonging is understood in the context of a diverse undergraduate population. Often developing a sense of belonging is assumed to be a universal and uniform experience rather than a classed, racialized and gendered experience [47]. Gourlay (2015) rightfully argues that belonging should not be understood as a normative category that can potentially be exclusionary and even harmful. He argues that appeals to "belonging" and "participation" can create structural distinctions and inequalities among different students that are discordant with the "day-to-day unfolding of higher education as situated social practice" [48] (p. 402). A second issue concerning all these discussions is the fact that the dominant discourses surrounding belonging often fail to consider a space for those students who may actively choose not to belong. In recent work, Gravett and Winstone (2019) explore how students' diverse backgrounds and specific interests may lead them to actively identify themselves as "outside the bubble" of dominant university communities [49] (p. 7). The complexities of belonging have been examined by Guyotte and colleagues (2019), who contend that for some students, un-belonging may be a positive, active, choice where students construct their identities in relation to their experiences: "belongingness is not inherently positive" [50] (p. 14). Thus, the critical literature on belonging and inclusion tell us that for multiple reasons, some students may resist the institutional or normative conceptions of belonging and community. Informed by all these discussions and debates, in this study, we wanted to better understand these diverging mechanisms in terms of belonging in online classrooms for minoritized students. Do they benefit from online education in terms of belonging in safety, or does the move to online education block their sense of belonging? What can we deduct then in terms of inclusive education debates? We use a recent case of online education during the COVID-19 pandemic at the Vrije Universiteit Amsterdam, which resonates with developments that took place in many higher education institutions all over the world.

## 3. Context and Methods

Our research took place at the VU in Amsterdam. The VU is one of the most ethnically diverse research universities in the Netherlands [28] and presents itself as a university that is committed to creating an open environment for a diverse group of students. One of the concrete initiatives is the development and rollout of the Mixed Classroom Educational Model, which capitalizes on differences between students and teaches students to take into account different perspectives [43].

Over the course of three months, in the spring of 2021, 27 semi-structured interviews were conducted with students who are minoritized based on various identity dimensions. The use of this qualitative approach fitted our interpretive research aims, namely gaining insight into students' experience in online synchronous classrooms. The respondents were found through the networks of the interviewers and organizational networks at the VU.

Based on the literature, we formulated the interview topics. The main topic of "sense of belonging" consisted of subthemes, including instructor–student interaction, student–peer interaction and course content applicability as well as feelings of safety in the classroom [20,23]. The prepared topics were followed by detailed probes tailored to the

interviewees' experiences, aiming to understand what affected their sense of belonging and the effects of their (diminished/enhanced) sense of belonging.

Because possible interviewees were identified through personal networks (socially present students) and student associations (socially active students) or because they contacted the researchers themselves when they heard of the project (proactive students) the results could be biased, as all students that were identifiable and willing to participate showed at least some levels of engagement. This means that our results, which are based on these relatively engaged students, probably represent a more engaged student population than average. This implies that for many students, their educational experiences during COVID-19 might have been even worse in terms of engagement and belonging.

We selected students who were studying at the VU at the time of the interview and had one or more minoritized student identity dimensions marker. Our sample consisted of students with the following minoritized identity dimensions: students with a physical impairment ($n = 1$); attention-deficit/hyperactivity disorder (ADHD) ($n = 2$); autism spectrum disorder ($n = 1$); a non-Western or Japanese migration background ($n = 20$); international student ($n = 4$, who were all white); and/or identify as LGBTQI+ ($n = 7$). We deliberately included interviewees with visible (for instance, non-western migration background) and non-visible (for instance, ADHD) identity markers.

Respondents were informed about the aim of our research and given transparency on the data management. Before the interview, an informed consent form was shared with the respondents via email. At the beginning of the interview, respondents were again orally informed of the contents of the consent form and were asked for their consent. To ensure anonymity and treat the data with care, the researchers used passwords on their computers, erased the recordings after transcription and anonymized the respondents in the transcript phase. The interviewers transcribed the interview in its original language (either Dutch or English). Dutch quotation were translated into English for this publication.

## 4. Results: Varying Experiences of (Un-)Belonging

Online education through online classrooms changed how classroom participants interact. The interviews were conducted one year after the sudden move to online education due to the COVID-19 outbreak. In most programs at the VU, on-campus courses were replaced by online, synchronous classrooms that make use of the online video conferencing tool Zoom, sometimes supplemented by short online asynchronous presentations. In Zoom meetings, participants generally see up to 25 other classroom participants at the same time on their screen (if using a computer or laptop). If more participants are present, the user can move between the screens. Every participant is represented by a square with a display name that displays the camera image or is black (when the camera is switched off). When someone shares their screen (such as the teacher, for a PowerPoint presentation), a student only sees the speaker and three other participants. When someone speaks, this person is displayed in a central position. Next to the video gallery, there is also a text-based chat function. The organizer, "host", has the opportunity to divide participants temporarily in smaller meeting groups, i.e., "breakout groups". Many teachers regularly made use of "Breakout rooms", in which students were divided online into smaller meeting groups.

From our data about the student experiences of belonging in online classrooms, three patterns emerged. (1) The most dominant experience was that the lack of connection in online classrooms reduced the student's sense of belonging. (2) In a few instances, however, the lack of connection and the heightened anonymity of online classrooms led to an increased sense of belonging and safety in online classrooms. (3) However, not all students experienced the switch to online education as a major change. Those who also had experienced their on-campus education as individualized, had an equally low sense of belonging in online and offline classrooms. Surprisingly, as we will describe, a low(er) sense of belonging did not always influence study success.

### 4.1. Pattern 1: Lower Sense of Belonging in Online Classrooms

The interviewees' stories showed that the students most often experienced a lower sense of belonging in online classrooms than in offline classrooms. They experienced a lack of connection, which often led to lack of safety, although they would not explicitly label it that way.

### 4.1.1. Lack of Connection in Online Classrooms

Many interviewees reported that they felt no connection with the classroom community, as they hardly had any interactions with fellow students, inside or outside of the online classroom. They missed not being able to easily interact one-on-one, for instance to ask a quick question or give an encouraging smile. In large classrooms, students often had their cameras switched off, making it even more complicated to form social connections. Some interviewees described that they sometimes almost forgot that fellow students were present in the classroom. Students felt that a connection, or even the possibility of a connection, with fellow students was absent in most online classrooms. Resonating with the experiences of other students, a student who reported that they actively participated in class stated:

> It [online classroom] doesn't feel like the classroom to me. It just feels like it's me and the lecturer.
>
> (Interviewee 21)

Students mentioned that online classrooms also impacted their connection with the teacher. They felt they were more anonymous to the teacher as well, since, for the teacher, each student is just one of many small screens. Interviewees felt they were interchangeable to the teacher, as they felt unseen by teachers and had (almost) no sense of connection to the teacher. The following student explains the difference between "real presence", in which the teacher can address a student with his gaze, and online interactions:

> In physical classrooms, the teacher looks at you and asks things directly at you. In physical classrooms, you feel way more present and the teacher sees that, while in online classrooms you just feel like a number, quite anonymous [author's translation].
>
> (Interviewee 1)

The absent interaction in online classrooms was often accompanied by a low sense of belonging. Because students felt anonymous in online classrooms, they did not feel like they belonged to a student community. A student explained:

> [B]ecause it's [online classrooms] like anonymous, it's not like you don't belong but you can never really... people can never know what you look like or who you are. You don't have to even say anything. It's not like I feel like I don't belong, [but] I also don't feel like I particularly belong, either. You just kind of feel like you're there to do what you have to do. But as a person, you don't matter much because you're not going to interact.
>
> (Interviewee 23)

This student's explanation illustrates how she feels that she does not feel present as a person. She does not feel seen and does not even feel that trying to be seen will make a difference. She explains that this impacts her feelings of belonging and that it makes her refrain from active participation and interaction. Apparently, when other students and the teacher do not know who you are, it feels as if your personal insight does not matter much, which strongly reduces the willingness to participate.

The lack of interaction and connection in the classroom also impacted the connection students felt to the VU and their student identity. Furthermore, the fact that they were not physically at the campus or entering the university buildings weakened their connection with the institution and its community. Students were no longer able to meet (new) people on campus, for instance through informal on-campus activities. This made students sometimes no longer feel like "a student" at all. The weaker connection resulted in an even lower feeling of belonging and made their study experience less enjoyable. As a Master's student explained:

> First [previous years studying at the VU] I did have the feeling of being a VU-student because I came into contact with other people at the VU, other students. Then you have the, well, the student experience: that you indeed meet people on campus and that you study on campus. Now I don't have that with online education [author's translation].

> (Interviewee 2)

In addition, the fact that studying took up less of their daily occupation, made students feel they like were no longer "a student". They instead felt like they took an e-learning course at home. As another interviewee stated:

> I feel way less like a student because it's not a big part of my day. I just open my laptop for a while, I go to a virtual lecture and then I close it again. Afterward, I just continue with my life. So yeah, I don't feel like a student compared to before Corona.

> (Interviewee 22)

Clearly, these interviewees indicated that they no longer felt connected to the university or their student identity in online education during the pandemic.

> Although a lot of the interviewees indicated that they missed the interaction when it was absent and enjoyed online classrooms more when there was some level of interaction, most of them still refrained from active participation which would establish interaction with peers and teachers. They showed a more passive attitude: 'It is what it is'.

> (Interviewee 23)

### 4.1.2. Lack of Safety in Online Classrooms

The interviews show that the lack of interaction led to anonymity and a lack of belonging that is needed to feel safe to participate. Although the interviewees did not explicitly label the online classrooms as unsafe, how they describe their reluctance to participate shows that they do not experience their online classroom as an entirely safe environment.

Interviewees explained that they found it difficult to anticipate how the classroom would react to their participation. They were reluctant to participate as they found it hard to predict whether their argument would be respected and valued. Online, it is harder to predict whether a statement is part of the majority opinion or a minority opinion for instance, what the perspectives of their fellow students are or whether the rest of the classroom is willing to try and understand a statement/opinion. This can be even more important for freshmen because they are still less familiar with the norms of classroom discussions in higher education. A first-year international student who identifies as queer reflects on this weakened sense of safety in online classrooms:

> If you feel the security of the group because you know that people have liked your contributions before, then you know that when you say something and

somebody doesn't agree with it, that someone will help you with the argument. In online classes, you don't have that. Maybe a little from the people you already know, the people you trust, (. . .) But then you only decide to make an argument if you are confident enough about what you're saying and that you know you can defend it all by yourself.

(Interviewee 26)

This quotation indicates a low sense of belonging: if students truly felt that they belonged in the online classroom, they would trust their contributions to be valued or they would feel more confident about possible reactions to their contributions. To feel confident that your contributions are valued and respected is important for a sense of belonging [20,23].

One of the students described a course in which the teacher managed to create an open atmosphere in which students felt it was safe to share. The teacher set the example and shared personal experiences, reflected on their own opinions, explicitly asked about others' experiences and encouraged students to share their experiences.

Multiple interviewees indicated that they felt a higher sense of belonging in smaller online classrooms or in smaller discussion groups. In smaller groups, they more often felt like they got to know their fellow students and felt safer when sharing. In smaller classrooms, the barriers to participating in group discussions are lower and students have more responsibility for the group dynamics.

In short, most of the reports showed that the interviewed students experienced the switch to online education in negative ways. It decreased their belonging, safety and participation and simply made studying less enjoyable, which derived from the lack of communicational cues and social presence, as described in the literature [38,39]. In their examples, students did not refer to their own minoritized identity dimensions. Instead, they argued that online education was troubling for every student and not only for students with minoritized identity dimensions.

### 4.2. Pattern 2: Higher Sense of Belonging in Online Classrooms

Not all students experienced a weakened sense of belonging in online classrooms. Multiple interviewees assumed that the anonymity of online classrooms reduced stereotyping (as also argued in the literature [12,42]), although for most this did not increase their sense of belonging. Two students did indicate that the switch to online education increased their sense of belonging. Because many personal characteristics are invisible in online classrooms, they felt more autonomy to choose which aspects to disclose and which to not disclose. This autonomy in how they present themselves increased their willingness to participate, and they even participated more actively in online classrooms than they had in offline classrooms. One of these students, who is a trans man, explained the higher sense of safety that he felt in online classrooms.

And that made it difficult [in offline, on-campus classrooms]—that I had to explain everywhere: 'Yes, I am trans, and would you please address me as a man?' I kind of liked education online because of that. Nobody uses anything apart from 'he' anymore. (. . .) It's just easier to put pronouns after your name and hope that people just do it right, instead of having to always explain it in front of the entire class. (. . .) This gives me more confidence to speak up, I know that they will not misgender me when they talk about it after. [author's translation]

(Interviewee 28)

Unlike in on-campus classrooms, where this student had to explicitly point out to everybody that he was a man, in online classrooms, he did not feel compelled to mention his gender because his printed name and pronouns clearly indicated that he was a man.

Also, another interviewee, who is hard of hearing, experienced online classrooms as less stressful. In on-campus classrooms, she felt that she was perceived as different from her fellow classmates because of her hearing aids. In online classrooms, her hearing aids went unnoticed. In online classrooms, asking questions through the chat to fellow students, in cases when she missed anything the teacher had said, was also widely accepted. Furthermore, she did not have to look around to identify the person speaking because when you participate in an online session, every sound comes from one place. For these two students, the decreased visibility in online classrooms made them feel more confident in participating.

*4.3. Pattern 3: Low Sense of Belonging in Online and Offline Classrooms*

Not all students experienced belonging in offline and online classrooms differently. While some students missed the connection with teachers and fellow students in online classrooms, some students did not miss this connection because they had also not felt this connection in offline, on-campus classrooms. They had simply never felt "seen" in higher education; they experienced higher education as a fairly anonymous experience in general. They related this to the size of their program and classrooms (both large) and the individual responsibility they felt over their studies. A Master's student stated:

> In general, there is not much personal contact at university. In a secondary school that's way more of a thing. For me it went quite abrupt, so: direct from secondary school to university. You come from an environment where there is a coordinator that personally speaks to you once every while to see how you are doing and then you get here at the VU. Then it's just... you're just a number. Now you're a student number and done: that's it. If they look at the attendance sheet, they see their name, number, and done. (...) It's probably also too much to know everyone [author's translation].
>
> (Interviewee 12)

This lack of belonging was described in the same terms as the others described their experiences in online classrooms: they felt like a "number" and did not feel seen or recognized as a person by the VU or the teacher. These interviewees felt in charge of their studies, both in offline and online education, and showed no expectation of becoming less anonymous.

Lack of Brave Spaces and Deep Learning

When there is a lack of connection and belonging in the online classroom, it is difficult to imagine a shared goal—which is important for making classrooms into safe spaces, particularly diverse classrooms [29]. Although none of the interviewees outright stated feeling unsafe in online classrooms, many interviewees described that they did not feel safe enough to participate. Online classrooms did not function as "brave spaces": spaces in which participants not only felt safe (often conflated with comfort) but also dared to participate and articulate their opinions, insights and experiences [51]. As the term "brave space" implies: participation needs bravery because even if it feels safe and comfortable to attend a class, this does not mean that participation is comfortable and cannot not lead to friction. After all, deep learning and changing your dispositions comes with discomfort.

All students had the same goal, passing the course, but they did not describe this as a collective endeavor. For them, classroom participation did not clearly contribute to passing the course and they did not acknowledge the importance of classroom interactions. They often only attended online classrooms to gather the information necessary to pass the course—not to engage with the course or fellow participants. From their own experience, it was enough to pay attention and study individually to pass a course, so they did not feel the necessity to be "brave" and to actively participate in online classrooms. This attitude is illustrative of a "strategic approach to learning" [25]: learning in order to pass

the course not to gain a deeper understanding (or "deep learning"). This detachment was strengthened because students did not know each other. They, therefore, needed even more bravery for participation and did not feel a responsibility to contribute to a collective learning community.

## 5. Discussion

We used in-depth interviews with minoritized students at the VU Amsterdam to advance our understanding of belonging in online, synchronous classrooms during the COVID-19 pandemic. When their education suddenly moved from campus to an online setting, minoritized students found themselves attending classes via online video conferencing tools in their own living rooms or bedrooms. This impacted their feelings of belonging in various ways.

Resonating with other research on online education, our study shows that the move to an online synchronous classroom heightened students' feelings of anonymity and reduced their feelings of belonging. Because of the lack of in-person interactions and informal interactions, students felt that neither the teacher nor their fellow students really knew them. This lack of connection, in combination with unfamiliarity with online interactions, made it harder to predict how others would react to contributions to classroom discussions. For the majority of students, it reduced their feelings of belonging and connection with the institution, and they no longer felt "like a student". Paradoxically, the heightened anonymity was accompanied by hypervisibility. In online conferencing tools, such as Teams or Zoom, names are explicitly displayed for every participant to see. Furthermore, students who actively participated in class and contributed to classroom discussions felt extremely exposed because every classroom member can see the speaking student in a central, front-facing position (if cameras are switched on), with their name displayed. Although sitting in your own physical space, not being known by others, can feel safe, in general, anonymity and hypervisibility raise barriers to participation. This reluctance to participate shows that, although anonymity can provide a safe space in which nobody knows you, this does not mean the anonymous online classroom results in a "brave space" in which students dare to actively engage in online classrooms and voice their opinions, insights and experiences [51].

However, the interviews reveal that anonymity can also have beneficial effects on feelings of safety, belonging and engagement for students, specifically for minoritized students. In line with the argument of Blake (2000) and Sullivan (2002) [15,16], the interviewees expressed that they assumed there were fewer micro-aggressions (stereotyping) in online classrooms, although this in itself did not increase their sense of belonging. Two interviews, however, show how the anonymity of online classrooms can increase self-confidence and participation. The online invisibility of characteristics that are normally visible in offline situations can increase the autonomy to present oneself. It can increase the freedom to choose which identity aspects or characteristics (like transgender identity or a hearing aid) to articulate and which to leave unnoticed. This, in combination with participating from the safety of their home environments and having less intensive interactions with other students, can encourage students (particularly minoritized students) to more actively participate than they would in on-campus classes.

In short, although the anonymity of online classrooms can be beneficial for some, most students found studying far less enjoyable after the move to online education. Although, as other research shows, the reduced belonging and engagement also applies to "non-minoritized" students, a lack of interaction and belonging is even more pressing for minoritized students, who generally experience more exclusion and who are often less familiar with the norms and codes in higher education. For them, measures to strengthen belonging and engagement in online education are more urgent.

Contrary to the argument that belonging is key to study success, lower levels of belonging and engagement did not seem to impact the study success of the students. Despite their low levels of engagement during online classes, interviewees were confident

they were still on track. They felt they could still prepare well enough to pass their tests, even in more time-effective ways than before.

The individualized stance of students who refrain from active classroom participation and do not feel responsible for collective classroom dynamics, suggests that deep learning might not be their goal, as it is not required to pass their tests. It could also be the case that, as Gravett and Winstone (2019) argue, unbelonging can be an active choice in the case of minoritized students [49]. Those with a "strategic approach" to learning, who focus on passing tests but not on deep learning for whatever reason [25], might be relatively satisfied with the move to online education for its time efficiency and place flexibility.

However, as belonging, participation and interactions are crucial to developing critical thinking [43] and, therefore, deep learning, a lack of belonging might have implications for study success if the course assessment requires deep learning. These findings show that, despite the opportunities that the flexibility and anonymity of online education provide, there are serious challenges in creating inclusive online education in which every student experiences belonging and engagement.

The context of this research was the COVID-19 pandemic. The mechanisms that we revealed are important because the situation at Vrije Universiteit Amsterdam (where offline, on-campus education was moved to online education) applies to many other educational institutions, inside and outside the Netherlands. Furthermore, it seems that online and digital education is here to stay [18,52]. However, some of the aspects described are particular to the pandemic situation. The most important aspect is the intense sense of isolation and the all-encompassing lack of social contacts, since during that period, nearly all in-person social contacts, including those outside the university, came to a halt. Furthermore, in most educational institutions, online COVID-19 education was "Emergency Remote Teaching"—a mere transfer of offline pedagogical approaches to online settings, in which the core teaching method mostly consisted of synchronous class meetings. This research shows that online education requires different pedagogies and deliberate educational designs that explicitly deal with the lack of in-person interactions and the possible effects on belonging and engagement [14]. Here, traditional universities can learn from educational institutions like Open Universities, which have been offering online courses that are deliberately designed for online settings.

HE teachers have an important role in creating inclusive classrooms [53–58]. These inclusive learning environments are defined by a number of researchers [59,60] around the themes of (1) stimulating student involvement, (2) accountability for structure and clarity, (3) attributing to a culture of openness and (4) a continuous critical reflection on one's own actions and attitude. Based on our findings, we formulated several recommendations for HE teachers to make online classrooms more inclusive, so that every student can experience belonging, resulting in deep levels of engagement. From our findings, we conclude that the accountability for structure and clarity and the culture of openness requires different actions from HE teachers in online classrooms than in physical classrooms. First, it is essential to create an open atmosphere in which students feel like the teacher is a person [61]. Teachers can achieve this by being honest and open about their own personal experiences, knowledge gaps and insecurities [43,52]. Second, it is important to deliberately organize opportunities for students to get to know each other. Because of the barriers to online interactions and the absence of spontaneously occurring informal interactions in purely online courses, this is even more important in online education [52,62]. Introduction activities allow students to get to know each other and the teacher and help create a safer—or rather, braver—classroom climate, which also stimulates student involvement. Working in small classrooms or small assignment groups makes it easier for students to get to know each other [52]. Teachers can also set time apart for students to get to know both each other and the teacher in a semi-informal atmosphere, which, in online settings, requires more deliberate initiative. Higher education institutions should give teachers the space to build these connections [54,63]. Third, explicating guidelines, rules and expectations reduce insecurity. It provides guidance and clarity when teachers communicate their expectations to students [18,53,64–67]. For

example, it stimulates participation and student involvement to explicitly mention that input is appreciated. Even when this feels self-evident to the teacher, this often is not the case for every student. Teachers can also have the group themselves formulate behavioral guidelines for discussions [67].

Lastly, we like to emphasize that although teachers are crucial actors for creating inclusive classrooms, their efforts remain isolated without the engagement of others. Creating inclusion in education requires the close involvement of teachers and students but also of the management and other faculty members. It requires the dedication of the entire institution [68,69]. We believe that promoting inclusive spaces in the higher education system requires better understanding of the nuanced, situated and contextualized accounts of students' belonging in the learning time–spaces of the university, including non-institutional spaces. We also believe that more critical research should be carried out in the literature in terms of the sense of belonging so that we do not forget to question who can belong as well as how and to where/whom. In a similar vein, we must also examine why minoritized students may choose to resist institutional or normative conceptions of belonging and prefer to form alternative connections and networks. We think that discussions on belonging in the context of higher education and minoritized students are very fruitful for bringing about novel perspectives of inclusion in education.

**Author Contributions:** Conceptualization, T.K.A., M.S. and S.M.-Y.; methodology, T.K.A.; writing—original draft preparation, T.K.A. and S.M.-Y.; writing—review and editing, T.K.A., S.M.-Y. and M.S.; supervision, M.S. and S.M.-Y. All authors have read and agreed to the published version of the manuscript.

**Funding:** This research received no external funding.

**Institutional Review Board Statement:** According to the research code of conducts of the FSW Research Ethics Review Committee (RERC) that entailed a self-check, it was suggested that ethical approval was not required for the type of study reported in the article.

**Informed Consent Statement:** Informed consent was obtained from all subjects involved in the study.

**Data Availability Statement:** Data is unavailable due to privacy.

**Conflicts of Interest:** The authors declare no conflict of interest.

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
