# Peer review of "Belonging in Online Synchronous Classrooms: Experiences of Minoritized Students in Dutch Higher Education during the COVID-19 Pandemic"

_education, doi:10.3390/educsci13100986_

Round 1
Reviewer 1 Report
There have been a lot of publications on the topic of online learning, belonging and engagement as a result of the swift pivot as a result of the pandemic, indeed these were topics of interest to researchers prior to 2020 and so there is much primary data and literature out there to utilise in a paper. It is of interest as the participant population is atypical although there is a widespread of non-standard students included in this very small scale study.
The first sentence of the introduction actually implies that some university campuses have stayed shut or not reopened as a result of the pandemic. Is this correct? The final sentence of the initial paragraph of the introduction also needs to be reviewed in terms of the tense.
I question the need for Section 2 ‘theory the relevance of belonging in online education’ at the start of the paper rather than integrated into the results or the discussion. It is rather long section and is not written succinctly. Indeed, this and the preceding section do not flow in terms of language style which needs to be revised, as at present it seems as though one or more people has written independent paragraphs and has been no segue between them.
In the results, the authors described several 'patterns' although a pattern infers something that has been measured several times and is seen to occur every time. I think that the authors have identified themes from the participants rather than a pattern here and the title should be changed. The authors quote several first hand comments in a variety of styles which need to be consolidated. Theme two (the increased belonging whilst online) has been identified from only two students so it is quite tenuous to draw conclusion. Finally theme free (sense of belonging) doesn't seem to relate to the quote provided which could be considered more to do with transition to higher education.
The authors continue to identify a lack of safety in online classrooms although potentially students or participants may have been feeling a lack of confidence? Indeed, lack of engagement doesn't necessarily mean they were feeling unsafe.
The paper could be improved by the addition of demographic data on the subjects involved such as the level of study at the point of online learning what type of delivery they were experiencing at the time of not only COVID but pre COVID and how long after a return to on campus delivery was this study undertaken.
Participants are very diverse and heterogeneous population they’re self-selecting and very small in number, one or two participants are associated with each type of minoritization. It would be useful to have some demographic data about the institution as a whole such as the percentage of minoritized students at the institution (does it have a widening participation remit?), the background of the students such as home environments, the access they had to good Internet access and quiet spaces etc. during the pandemic in which to do their work.
Overall, the paper needs a fair a mount of revision in terms of grammar and in text citations. I appreciate that English may not be the first language of the authors and that they may have used a translation service to produce the document but in parts sentence structure is not good. On occasion this might be missing a small word such as ‘a’ or ‘an’ at other times there's an unusual word utilised in the sentence structure which may or may not have come from the use of a thesaurus, an example which is the use of ‘unsafety’ in line 147 and ‘deduct’ rather than ‘deduce’ in line 182.
At times it is very hard to comprehend the meaning of a sentence for example line at 278 interviewees felt they 'were interchangeable to the teacher' and in line 287 'the absent interaction' don't quite make sense. These sentences would benefit from a revision of sentence structure.
On occasion the authors contradict themselves such as the paragraph that starts line 215 seems to be contradicted by the paragraph that starts line 223, in showing the anonymity of the subjects.
The authors need to make sure that their in text citations and reference list adhere to the journals policy as at present they do not. For example, line 139 where the authors have written ‘and’ between the two references they need to determine if this is what the general requires.
Author Response
Dear Reviewer,
Thank you very much for your valuable input. We believe that changes in reaction to the comments have lifted the manuscript to a higher level. In reaction to your remarks below we explain the argumentation for changing or not changing the text and the ways in which the text was changed.
Overall, additional information has been added to further clarify our approach and results, this includes but is not limited to the selection of participants in the study, the analyses methods, and the use of concepts such as ‘lack of emotional safety’. Your remarks on language have also been consolidated and in addition, a native speaker has edited the manuscript.
We numbered your remarks and below we will try to address these; Thank you very much again for helping us out with your valuable insight.
1.1. The first sentence of the introduction actually implies that some university campuses have stayed shut or not reopened as a result of the pandemic. Is this correct? The final sentence of the initial paragraph of the introduction also needs to be reviewed in terms of the tense.
It was not the intent to insinuate the definite closure of university campuses and we are unaware of instances where this is the case. Upon reread we do understand how this can be interpreted this way as Reviewer 1 states. For clarification of our statement we changed ‘closure’ to ‘temporary closure’. In the final sentence of the first paragraph, the present tense has been changed to past tense, as well as adding ‘and the resulting digitalization of higher education’ to showcase the enduring need of researching online and hybrid classrooms as well as being aware of the ending of the situation as it was during the peak of Covid-19.
- Added ‘temporary’ to first sentence. (p. 2); Added ‘and the resulting digitalization of higher education’ to final sentence of paragraph. (p.2)
1.2. I question the need for Section 2 ‘theory the relevance of belonging in online education’ at the start of the paper rather than integrated into the results or the discussion. It is rather long section and is not written succinctly. Indeed, this and the preceding section do not flow in terms of language style which needs to be revised, as at present it seems as though one or more people has written independent paragraphs and has been no segue between them.
We reviewed the remark on the relevance of section 2, the position in the paper it is placed, and its writing style in comparison to the introduction. After review of the options (pros and cons) of integrating this section within other headings of the journal, we decided to shorten the section massively and bring it down to the original intent of the section instead of integrating it solely within the results section. Reasons for this decision were: 1. We believe it necessary for the reader to have a background knowledge of the main concepts with a theoretical base, (which cannot be included in the introduction thoroughly enough due to point 2) and 2. the main debates surrounding these concepts, which due to the sensitive nature of the subject matter are important to treat with care and be aware of while being presented with the results. If not, the reader cannot trust that we have come to our outcomes with outmost care for not only the participants but with care for effects for the greater student population. We do acknowledge the length of section 2 and shortened it, while also rewriting it to match the style of the rest of the manuscript, to reach these two aims more succinctly.
- The section (2. Theory: the Relevance of Belonging in Online Education) has been shortened and rewritten.
1.3. In the results, the authors described several ‘patterns’ although a pattern infers something that has been measured several times and is seen to occur every time. I think that the authors have identified themes from the participants rather than a pattern here and the title should be changed. The authors quote several first hand comments in a variety of styles which need to be consolidated. Theme two (the increased belonging whilst online) has been identified from only two students so it is quite tenuous to draw conclusion.
We understand your remarks on the usage of pattern and we agree with the remark. Therefore, we changed ‘pattern’ in the title to ‘experience’ as we believed this was more appropriate to the content discussed than ‘theme’. Hereby we also considered the remark on the tenuous nature of pattern/theme/experience two. Due to the impression that the participants are generally more active and connected students (as stated in the method section), we believe this experience might be more widely shared among the student population for those students who are more passive and less connected. This sense is increased due to this experience being held by a student who is hard of hearing and a trans man, whose identity characteristics are not as well-documented in this study as others (for instance non-western migration background). This connects to a later comment of yours on the selection of participants.
- We have added two sentences to the method section stating: ‘The aim was to explore what the experiences of minoritized students during online education during COVID-19 were. Although qualitative research cannot be generalizable to large populations, we believe that the findings of this study are transferrable to other students and contexts.’
1.3. Finally theme free (sense of belonging) doesn't seem to relate to the quote provided which could be considered more to do with transition to higher education.
Experience 3 is on the low sense of belonging in higher education, both in online higher education and offline higher education. The quote used in ‘Experience 3’ illustrates the sense of anonymity in higher education, higher education when courses are taught online or offline. We shortened the quote to more clearly illustrate the important aspect of the quote that illustrates Experience 3 as its usage was not entirely clear in the original form as you stated.
- Reevaluated the quote in Experience 3 and removed an aspect to not distract from the illustration to low sense of belonging in online and offline higher education that it makes.
1.4.The authors continue to identify a lack of safety in online classrooms although potentially students or participants may have been feeling a lack of confidence? Indeed, lack of engagement doesn't necessarily mean they were feeling unsafe.
Both you and the other reviewer questioned the use of ‘lack of safety’ in our results section (see also 2.6.). Hereafter we reconsidered the usage of the term and reviewed the data and the conclusions made from that data. We understand the comments and consolidated with our aims in the following ways: by adding a table on the code tree used to code and then analyze the interviews we added a column ‘notes’ in which the code (rows) is elaborated upon. For the code connected to ‘lack of safety’ we changed to ‘lack of emotional safety’ as we intended the phrase to be understood. To explain the concept of emotional safety we explained the concept in the notes column and added references to showcase the basis of the use of safety as we employ it and its widely usage. Throughout the text, we changed safety to emotional safety where deemed necessary and evaluated the conclusion of lack of emotional safety. Hereby we changed the wording clearly stating the researchers’ role in identifying the theme (in accordance with the remark by reviewer 2) and so added nuances.
- Example of changes made to the text:) The interviewees’ stories showed that the students experienced a lower sense of belonging more often in online classrooms than in offline classrooms. They experienced a lack of connection, a lack of social presence, which made them refrain from active participation for reasons that resonate with what is called a lack of ‘emotional safety’ (see Table 2. and for more information: Harless, 2018; [removed]).B.) The interviews show that the lack of interaction led to anonymity and a lack of belonging that is needed to feel safe enough to feel brave (Palfrey and Ibargüen, 2014), e.g., feel confident enough to participate. Although the interviewees did not explicitly label the online classrooms as unsafe, possibly due to the perception of “safety” as being limited to “physical safety” and not including of ‘emotional safety’ (see Quiros, Kay, and Montijo, 2013; Shean and Mander, 2020; Ambrose et al. 2023; [removed]), how they describe their reluctance to participate and fear to the reaction of others to participation resonates with how emotional unsafety is understood in the literature.
1.5. The paper could be improved by the addition of demographic data on the subjects involved such as the level of study at the point of online learning what type of delivery they were experiencing at the time of not only COVID but pre COVID and how long after a return to on campus delivery was this study undertaken.
The phase of education: bachelor/masters has been added to the text. Here, an extra distinction is made between first-year students and master students who did their bachelors at a different university. This to showcase the included experience of those who are new to the university where our research took place and can thus be expected to feel less belonging at the physical campus.
The interviews were taken during the closure of the campus, namely in the spring of 2021, as stated in the methods section.
- After: ‘Over the course of three months, in the spring of 2021, 27 semi-structured interviews were conducted with students who are minoritized based on various identity dimensions.’ Added: ‘At the moment of closing down the university campus, 12 of these students were in their masters, 10 of which had followed their bachelors at the same university, and 15 in their bachelors, of which 2 first-years. Their study programs are divided among 6 faculties: -science, -humanities, -law, -social sciences, -business and economics, -behavioral and movement sciences.’
1.6.Participants are very diverse and heterogeneous population they’re self-selecting and very small in number, one or two participants are associated with each type of minoritization. It would be useful to have some demographic data about the institution as a whole such as the percentage of minoritized students at the institution (does it have a widening participation remit?), the background of the students such as home environments, the access they had to good Internet access and quiet spaces etc. during the pandemic in which to do their work.
In reaction to your comments we made changes to the method section connecting to the participants: Besides the addition of the available data on the higher education institution’s student population, which is minimal in accordance with general policy of the Netherlands, we gave additional information on the participants and how we reached them.
- Expanded the Context and Methods section to include more information on how participants were reached. The added aspects were:
- The use of this qualitative approach fitted our interpretive research aims, namely, gaining insight into students’ experience in online synchronous classrooms. The participants were found using the interviewers’ networks with teacher colleagues who shared the call for participants on the online campus platform Canvas, using snowballing by contacting students who shared the call with other students in their year, and using organizational student networks for minoritized students at [removed], such as [removed] and [removed]. We chose the snowballing technique due to wanting to reach students with specific identity characteristics who can not be selected using administrative data. The aim was to explore what the experiences of minoritized students during online education during COVID-19 were. Although qualitative research cannot be generalizable to large populations, we believe that the findings of this study are transferrable to other students and contexts. ‘Because possible interviewees were identified through personal networks (socially present students), student associations (socially active students), or because they contacted the researchers themselves when they heard of the project through their online classroom platform (pro-active students).’
- ‘We selected students who were studying at [REMOVED] at the time of the interview and had one or more minoritized student identity dimensions markers. We specifically advertised for students with one or more minoritized identity dimensions markers when contacting student associations, personal connections, and in the call shared on online classroom platforms.’
Reviewer 2 Report
Overall, I believe the article addresses a timely important topic in today’s (and tomorrow’s) higher education landscape. However, I have some concerns that need clarified with the paper’s methodology (summarized in the “major issues” section below) before I can recommend it for publication. Additionally, several minor concerns abound, including numerous grammar issues, a few of which I have pointed out in the “minor issues” section below. I recommend an additional one or two native or expert academic English proofreaders for this paper.
Major:
Context and Methods: Include the interview questions.
Recruitment information, "The respondents were found through...:" How? Through personal social media posts? From speaking to your students, friends, etc.? Need to be more specific about recruitment. This is known as "snowball sampling" when you ask participants to help you find more participants, and you need to acknowledge its limitations and justify its use.
Page 5, line 209, "heard of the project:" Heard of it how? Was it advertised somewhere? Or do you just mean by word-of-mouth from students you had reached out to directly?
Page 5, line 216, regarding participant minoritized identity dimensions: How did you know this about them? Did you do a screening questionnaire? Did you advertise specifically for people of certain groups? Etc.
Results: Before this, some description of what method you used to analyze the responses and ensure trustworthiness, etc. need to be reported. In quantitative work we would be a lot more detailed about our reliability and validity of measurements, as well as a data analytic strategy, and the same rigor should be required for qualitative work. I find it interesting I didn’t even see “content analysis” and “themes” mention, as I would expect in qualitative interviews like this one.
Page 6, line 261, "although they would not explicitly label it that way.": This is doing a lot of heavy lifting in your entire theme grouping and I'm finding myself wanting far more clarification. Maybe something like "which often led to a theme identified by the researchers as "lack of safety."" I'm also unsure you should "put words in [participants'] mouths" in this way. Maybe you could label the phenomena you found in a different way, more consistent with participants’ words.
Recommendations section of Discussion, beginning with "Based on our findings, we formulate several recommendations..." These recommendations are not novel discoveries of this paper, so they should probably be cited. For example, your "teacher as a person" concept seems to touch on the different types of "presence" (like social, teaching, etc.) a teacher can have in a course (including online). The rest of your points seem to be about how an instructor can achieve the different types of presence in an online course. You may find the following sources helpful to cite to back up these practices already recommended by others.
Bromer, Billi. (2022). "The Importance of Social Presence in Online Instruction." The Academic Minute. WAMC. https://academicminute.org/2022/01/billi-bromer-brenau-university-the-importance-of-social-presence-in-online-instruction/
d’Alessio, Matthew A., Schwartz, Joshua J.. Pedone, Vicki, Pavia, Jenni, Fleck, Jenna, & Lundquist, Loraine. (2019) "Social presence enhances student performance in an online geology course but depends on instructor facilitation." Journal of Geoscience Education 67:3, pages 222-236.
Garrison, D. R., Anderson, T., & Archer, W. (2000). Critical Inquiry in a Text-Based Environment: Computer Conferencing in Higher Education. The Internet and Higher Education, 2, 87-105. http://dx.doi.org/10.1016/S1096-7516(00)00016-6
Hanstedt, Paul. (2018). "Creating Wicked Students: Designing Courses for a Complex World." Stylus Publishing LLC.
Hillman, D., Schudy, R., & Temkin, A. (2020). "Best practices for administering online programs." Taylor & Francis Group. (Chapter 6)
Martin, Florence & Wang, Chuang & Sadaf, Ayesha. (2018). "Student perception of helpfulness of facilitation strategies that enhance instructor presence, connectedness, engagement and learning in online courses." The Internet and Higher Education, Volume 37, Pages 52-65.
University of Minnesota. (2022). "Active Learning." Retrieved from https://cei.umn.edu/teaching-resources/active-learning
Minor issues:
Your title should have a semi-colon instead of a period between “Classrooms” and “Experiences.”
Page 1, line 40 "attainment": I think "achievement" works better here.
Page 2, sentence beginning with "With inclusive classrooms...": Confusing wording and not grammatical. Example of how to fix: "Inclusive classrooms are those in which every..."
Page 2, line 65: Unnecessary comma.
Page 2, line 68 semi-colon: This should probably be a comma instead, or perhaps the question should be set off in quotation marks.
Page 2, line 79, "knowledge clips:" What does this mean?
Page 2, line 95 "likelier:" Should be more likely.
Page 3, line 104, "Minoritized students,": Unnecessary comma.
Page 3, line 116, "equity not...": Should be "instead of" in place of "not."
Page 3, line 127: Use "is" instead of "are," as it refers to singular "lack of student engagement."
Page 4, line 196, "fitted:" "Fit" is the more appropriate simple past tense form here. "Fitted" is more appropriate for when something "has been fitted," like clothing being tailored or a model being "fitted" to data.
Page 4, line 202: You have two "ands" in this series.
Page 5, line 246, "Breakoutrooms:" This should be separated into two words.
Page 6, line 273, "who said to actively participate:" What does this mean? Said they actively participated? Said they did not? Were told to actively participate? Etc.
Page 6, line 273: "Told" needs a direct object. We say "said" instead in these cases.
Page 8, line 385, "transman." This should be separated into two words. Using "trans" as just another out of many possible adjectives that could "man" normalizes transgender identity. Compounding the words unnecessarily "others" trans people by labeling them as something other than "man" or "woman" (even if their identity is binary man, for example). Furthermore, the combination of the words into one is commonly done by gender-critical (anti-trans) activists (for the "othering" reason stated above), so inclusive writing should avoid this association.
Page 8, line 440, "your:" You've used this informal "you" a couple of times: use third-person language instead, e.g., "people" and "they" do [this and that] instead of "you" [verb]."
Page 10, line 465: The first "nor" should be "neither."
Page 10, line 465, "Teacher, nor their fellow students, really..." I don't think any commas are needed here. In general, you overuse commas.
Page 11, line 507: "and feel not responsible:" This should be "do not feel responsible."
Page 11, line 508, sentence beginning with "or." Avoid beginning sentences with coordinating conjunctions (and, or, but, for, yet, nor).
Page 11, line 508, sentence beginning with "or." The common wording is "could well be the case," but I discourage this wording. Try something like, "However, perhaps unbelonging can be an active choice among minoritized students, as Gravett and Winstone (2019) argue," for example.
Page 12, line 559, "the management and other faculty members.": In a USA context, we might say "administration, staff, and other faculty." We don't call college employees "management," typically. I don't think I see that term used for UK unis either. I'm not sure if this is merely a wording/translation issue, or whether the structure of Dutch universities is very different from what I know as an American. I would be careful with this wording and check with someone else who is familiar with both Dutch universities and the common US and UK English usage for employee types at universities, if possible.
Grammar issue on page 12, line 565, "we have to then dive into as to why..." Maybe simpler wording like just "investigate" would suffice here.
References: The spacing, typeface, and indenting of the reference formatting are messy.
Regarding grammar issues, the list above is not exhaustive, but instead representative of common mistakes found throughout the manuscript.
Author Response
Dear Reviewer,
Thank you for your attentive reads and constructive comments. We believe the changes in reaction to your comments have lifted the manuscript to a higher level. In reaction to your remarks we explain the argumentation for changing or not changing the text and the ways in which the text was changed. Overall, additional information has been added to further clarify our approach and results, this includes but is not limited to the selection of participants in the study, the analyses methods, and the use of concepts such as ‘lack of emotional safety’. The remarks on language have also been consolidated and in addition, a native speaker has edited the manuscript. Here again, we would like to thank you for helping us to improve our paper. Below we would like to go over your comments one by one.
2.1. Context and Methods: Include the interview questions.
In response to your comment 2 the topics discussed in the interviews the interview topic list has been added in Table 1. Interview Topic List.
2.2. Recruitment information, "The respondents were found through...:" How? Through personal social media posts? From speaking to your students, friends, etc.? Need to be more specific about recruitment. This is known as "snowball sampling" when you ask participants to help you find more participants, and you need to acknowledge its limitations and justify its use.
2.3. Page 5, line 209, "heard of the project:" Heard of it how? Was it advertised somewhere? Or do you just mean by word-of-mouth from students you had reached out to directly?
2.4. Page 5, line 216, regarding participant minoritized identity dimensions: How did you know this about them? Did you do a screening questionnaire? Did you advertise specifically for people of certain groups? Etc.
In relation to all these remarks numbered 2.2., 2.3., 2.4., which connect to remarks made by the other reviewer, we made changes to the method section connecting to the participants: Besides the addition of the available data on the higher education institution’s student population, which is minimal in accordance with general policy of the Netherlands, we gave additional information on the participants and how we reached them.
Expanded the Context and Methods section to include more information on how participants were reached. The added aspects were:
-
- The use of this qualitative approach fitted our interpretive research aims, namely, gaining insight into students’ experience in online synchronous classrooms. The participants were found using the interviewers’ networks with teacher colleagues who shared the call for participants on the online campus platform Canvas, using snowballing by contacting students who shared the call with other students in their year, and using organizational student networks for minoritized students at [removed], such as [removed] and [removed]. We chose the snowballing technique due to wanting to reach students with specific identity characteristics who can not be selected using administrative data. The aim was to explore what the experiences of minoritized students during online education during COVID-19 were. Although qualitative research cannot be generalizable to large populations, we believe that the findings of this study are transferrable to other students and contexts. ‘Because possible interviewees were identified through personal networks (socially present students), student associations (socially active students), or because they contacted the researchers themselves when they heard of the project through their online classroom platform (pro-active students).’
‘We selected students who were studying at [REMOVED] at the time of the interview and had one or more minoritized student identity dimensions markers. We specifically advertised for students with one or more minoritized identity dimensions markers when contacting student associations, personal connections, and in the call shared on online classroom platforms
2.5. Results: Before this, some description of what method you used to analyze the responses and ensure trustworthiness, etc. need to be reported. In quantitative work we would be a lot more detailed about our reliability and validity of measurements, as well as a data analytic strategy, and the same rigor should be required for qualitative work. I find it interesting I didn’t even see “content analysis” and “themes” mention, as I would expect in qualitative interviews like this one.
After rereading the methods section, we agree with you on the lack of information available. To give more extensive information on the methods used a table of the codes has been added (see Table 2. Code groups). The table is accompanied with short description of the analyses in the text.
- Table on the codes and information has been added to the text: ‘Participants were informed about the aim of our research and about how their data would be managed. The interviews were over Zoom and lasted between 45 and 90 minutes. Before the interview, an informed consent form was shared with the participant through email. At the beginning of the interview, participants were again orally informed of the contents of the consent form and were asked for their consent. To ensure anonymity and treat the data with care, the researchers used passwords on their computers, erased the recordings after transcription, and anonymized the participants in the transcript phase. The interviewers transcribed each interview in its original language (either Dutch or English). Dutch quotes were translated into English for this publication. The transcripts were thematically coded based on the existing literature using a two-phase coding process that combined a deductive and inductive approach: the deductive code-groups were determined beforehand and applied in the first phase, and the inductive sub-codes during the second phase (see Table 2. Code groups for the coding table).’
2.6. Page 6, line 261, “although they would not explicitly label it that way.”: This is doing a lot of heavy lifting in your entire theme grouping and I’m finding myself wanting far more clarification. Maybe something like “which often led to a theme identified by the researchers as “lack of safety”. I’m also unsure you should “put words in [participants’] mouths” in this way. Maybe you could label the phenomena you found in a different way, more consistent with participants’ words.
Remark 2.6. connects to a remark made by again the first reviewer. We made necessary changes.
You rightfully questioned the use of ‘lack of safety’ in our results section (see also 2.6.). Hereafter we reconsidered the usage of the term and reviewed the data and the conclusions made from that data. We understand the comments and consolidated with our aims in the following ways: by adding a table on the code tree used to code and then analyze the interviews we added a column ‘notes’ in which the code (rows) is elaborated upon. For the code connected to ‘lack of safety’ we changed to ‘lack of emotional safety’ as we intended the phrase to be understood. To explain the concept of emotional safety we explained the concept in the notes column and added references to showcase the basis of the use of safety as we employ it and its widely usage. Throughout the text, we changed safety to emotional safety where deemed necessary and evaluated the conclusion of lack of emotional safety. Hereby we changed the wording clearly stating the researchers’ role in identifying the theme (in accordance with your remakr) and so added nuances.
- Example of changes made to the text:) The interviewees’ stories showed that the students experienced a lower sense of belonging more often in online classrooms than in offline classrooms. They experienced a lack of connection, a lack of social presence, which made them refrain from active participation for reasons that resonate with what is called a lack of ‘emotional safety’ (see Table 2. and for more information: Harless, 2018; [removed]).B.) The interviews show that the lack of interaction led to anonymity and a lack of belonging that is needed to feel safe enough to feel brave (Palfrey and Ibargüen, 2014), e.g., feel confident enough to participate. Although the interviewees did not explicitly label the online classrooms as unsafe, possibly due to the perception of “safety” as being limited to “physical safety” and not including of ‘emotional safety’ (see Quiros, Kay, and Montijo, 2013; Shean and Mander, 2020; Ambrose et al. 2023; [removed]), how they describe their reluctance to participate and fear to the reaction of others to participation resonates with how emotional unsafety is understood in the literature.
2.7. Recommendations section of Discussion, beginning with "Based on our findings, we formulate several recommendations..." These recommendations are not novel discoveries of this paper, so they should probably be cited. For example, your "teacher as a person" concept seems to touch on the different types of "presence" (like social, teaching, etc.) a teacher can have in a course (including online). The rest of your points seem to be about how an instructor can achieve the different types of presence in an online course. You may find the following sources helpful to cite to back up these practices already recommended by others.
Upon considering this remark by you, we agreed that rigidity here could be strengthened. The articles shared by you were read as well (re)reading articles on the topic through our own search. After which we evaluated the discussion.
- To ensure rigidity in the discussion, sources have been added to this section . In addition, some changes have been made to the text:
- ‘Based on our findings, in line with and in addition to the existing (gray) literature (such as the recently Erasmus+ project [removed]; authors et al.), we formulate several recommendations to make online classrooms more inclusive, so every student can experience belonging, resulting in deep levels of engagement..’
- ‘This links with the challenge of establishing social presence, or here ‘teacher presence’ in online classrooms (see, for instance, d’Alession et al. 2017; Garrison et al. 2003; Balboni et al. 2018; Rodríguez-Ardura & Meseguer-Artola 2016; Martin et al. 2020).’’
- ‘Introduction activities let students get to know each other and the teacher, and help create a safer – or rather, braver – classroom climate (for more ways to establish these moments of interaction, see Pacansky-Brock et al. 2020).’’
We would again like to thank you for your thorough reading of our paper to improve it; we believe that it has become much more academically sound due to your valuable insights.
Warm regards,
The authors of ‘Belonging in Online Synchronous Classrooms. Experiences of minoritized students in Dutch higher education during the COVID pandemic’

Round 2
Reviewer 2 Report
My previous issues have been addressed.
Author Response
no more comments from the reviewer.